# Comparative Analysis of Flavor Quality of Beef with Tangerine Peel Reheated by Stir-Frying, Steaming and Microwave

**DOI:** 10.3390/foods14173017

**Published:** 2025-08-28

**Authors:** Kaixian Zhu, Huaitao Wang, Hongjun Chen, Wenzheng Zhu, Chunlu Qian, Jun Liu, Juan Kan, Man Zhang

**Affiliations:** 1Cuisine Science Key Laboratory of Sichuan Province, Sichuan Tourism University, Chengdu 610100, China; prkxzdsys@126.com; 2College of Food Science and Engineering, Yangzhou University, Yangzhou 225127, China; 13914142904@163.com (H.W.); chenhj0123@163.com (H.C.); clqian@yzu.edu.cn (C.Q.); junliu@yzu.edu.cn (J.L.); kanjuan@yzu.edu.cn (J.K.); 3College of Tourism and Culinary, Yangzhou University, Yangzhou 225127, China; zhuwz@yzu.edu.cn; 4State Key Laboratory of Food Science and Resources, School of Food Science and Technology, Collaborative Innovation Center of Food Safety and Quality Control in Jiangsu Province, Jiangnan University, Wuxi 214122, China; 5Anhui Province Key Laboratory of Functional Compound Seasoning, Anhui Qiangwang Seasoning Food Co., Ltd., Fuyang 236500, China

**Keywords:** gas chromatography–mass spectrometry, gas chromatography–ion mobility spectrometry, odor activity value, reheating methods, taste

## Abstract

A prepared dish needs to be reheated before eating, and various reheating methods affect its flavor quality. This study evaluated the influence of stir-frying reheating, steaming reheating and microwave reheating on moisture content, lipid oxidation and flavor profiles of prepared beef with tangerine peel. Stir-frying reheating samples obtained a higher moisture content and the highest thiobarbituric acid reactive substance value. Fifty-seven volatile compounds were identified by gas chromatography–mass spectrometry, of which fifteen compounds were considered as odor-active compounds with an odor activity value > 1. Aldehydes were the most prominent contributors to the aroma of reheated samples. Results revealed that stir-frying reheating samples had the most varieties of odor-active compounds, and the odor activity values of most of them were relatively higher. The heatmap analysis based on the odor activity values indicated that the stir-frying reheating process could maintain the original flavor of samples. A total of fifty-two volatile organic compounds were identified by gas chromatography–ion mobility spectrometry, and the principal component analysis revealed that the three reheated samples could be well distinguished from each other. Moreover, the content of free amino acids and nucleotides in stir-frying reheating samples was higher than that in other reheated samples. In conclusion, different reheating treatments affected the flavor quality of beef samples, and stir-frying process was better to obtain the aroma and taste characteristics of samples. The results of this study could provide useful information about the appropriate reheating method of a dish of prepared beef with tangerine peel for consumers, caterers and industrial production.

## 1. Introduction

Prepared dishes are convenience foods that are easy for consumers to cook and consume. Based on the degree of convenience and processing levels, they are categorized into ready-to-eat, ready-to-cook and ready-to-heat food [1]. The prepared dish market has experienced rapid growth in recent years, especially in China, North America, Japan and Germany [2]. With the development of cold chain technology and the acceleration of the pace of modern life, the majority of consumers are inclined to consume prepared dishes instead of traditional home-cooked dishes and the varieties of prepared dishes become diverse. The prepared dishes first undergo a preheating or precooking process, and then they are vacuum- or atmosphere-packaged and stored in a frozen state and simply reheated before eating [3]. The prepared dishes have gained increasing popularity due to their convenience for consumers who would like immediate consumption with minimal reheating [4].

Beef with tangerine peel is a famous Sichuan dish and a well-known traditional meat product in China. The beef with tangerine peel dish is made with beef as the main ingredient, complemented by tangerine peel and a variety of other seasonings as auxiliary materials, which is distinguished by the unique aroma of tangerine peel [5]. In recent years, the beef with tangerine peel product has been present in the market in the form of frozen prepared food, which requires reheating before eating. This dish made as a prepared dish not only enriches the varieties of prepared dishes, but also provides consumers with a new option.

The common reheating treatments to cook meat products contain stir-frying and steaming in households and restaurants. The microwave oven is widely used due to its short heat time and low energy usage [6]. Different reheating treatments could lead to the changes of edible quality and flavor properties in meat products [7]. Li et al. compared the quality of a braised beef with potatoes dish reheated by open flame, microwave, steaming and boiling and found that open flame reheating method gained better edible quality and flavor characteristics of the beef, potatoes and soup [8]. Ping et al. evaluated the effects of different reheating treatments (microwave, boiling and steaming) on the flavor characteristics of Ceramic-Pot Sealed Meat and revealed that the microwave process was better to preserve the original flavor of Ceramic-Pot Sealed Meat [9]. Sheng et al. demonstrated that air-frying reheating method was conducive to producing more sulfur-containing compounds and pyrazines with a pleasant aroma compared with other reheating methods (boiling, baking, pan-frying and microwave) in prepared meat patties [10]. However, studies about the influence of reheating processes on the flavor profiles and edible quality of beef with tangerine peel have not been reported.

Flavor quality is one of the most important indicators for meat products, as it can affect consumers’ purchasing decisions and preference [11]. Gas chromatography–mass spectrometry (GC–MS) and gas chromatography–ion mobility spectrometry (GC–IMS) techniques are the commonly used methods to analyze the volatile compounds of meat products. GC–MS is used for the qualitative and quantitative analysis of flavor components, while GC–IMS can be applied for the identification of specific flavor substances and the comparative analysis of flavor differences in samples [12,13]. Moreover, the GC–IMS technique combines the high separation capability of GC with the high sensitivity of IMS and can be used for the rapid detection of volatiles. In recent years, the combination of GC–MS and GC–IMS has been able to provide a comprehensive flavor profile of meat products, which is important for analyzing the quality [14]. Pan et al. demonstrated that GC–MS could analyze a greater number of flavor substances in spiced beef. GC–IMS was more sensitive in detecting alcohols and ketones, and the combination of the two methods provided a complementary analysis of the flavor characteristics of spiced beef [15]. However, few studies had evaluated beef with tangerine peel prepared in different reheating methods using the combination of GC–MS and GC–IMS.

The objective of this study was to elucidate the changes in the flavor quality of the prepared beef with tangerine peel product during reheating and select the appropriate reheating process. Thus, this study assessed the effects of different reheating methods on the moisture content, lipid oxidation and flavor characteristics of beef with tangerine peel. The aroma profiles were evaluated by GC–MS, odor activity value and GC–IMS. The taste properties were analyzed by free amino acids and 5′-nucleotides. Moreover, sensory evaluation was also applied to investigate the quality of different reheated samples. The results could provide a theoretical foundation for the quality control and improvement of the prepared beef with tangerine peel dish and provide valuable information with potential application for other prepared meat products.

## 2. Materials and Methods

### 2.1. Materials and Chemicals

Fresh beef knuckle meat was purchased from the Sam’s Club in Yangzhou, Jiangsu province (China). Tangerine peel, salt, ginger, cooking rice wine, dried red pepper, Sichuan pepper, soy sauce, sugar and peanut oil were obtained from the RT-Mark in Yangzhou, Jiangsu province (China). The internal standards of 1,2-dichlorobenzene and n-alkane mixture (C_7_–C_40_) in chromatography grade were purchased from Sigma-Aldrich Co., Ltd. (Shanghai, China). Trichloroacetic acid, thiobarbituric acid and other reagents of analytical grade were obtained from Sinopharm Chemical Reagent Co., Ltd. (Shanghai, China).

### 2.2. Sample Preparation

#### 2.2.1. Preparation of Beef with Tangerine Peel

The beef knuckle meat was removed external connective tissue and washed with water, then cut into cubes (1.5 cm × 1.5 cm × 1.5 cm). Then, the beef knuckle samples (400 ± 5 g) were mixed and marinated with salt, ginger slices and cooking rice wine for 15 min and drained off to remove the surface juice excessed. The beef cubes were fried for 70 s at 180 °C in a deep fryer (LC-DZL05, Guangzhou Qingbang Kitchen Utensils Co., Ltd., Guangzhou, China) and then stir-fried with tangerine peel, dried red pepper and Sichuan pepper for 5 min (stir-frying rate of 10 times/min with a 6 s interval) in a pan heated by an electromagnetic oven (C22-RT22E01, Midea Group Co., Ltd., Shenzhen, China) at 800 W. Then the samples were stewed with water, cooking rice wine, soy sauce and sugar for 30 min at 1200 W, and finally the sauce was concentrated at 2200 W for 8 min. Nine batches of the above samples were prepared. After cooking, the samples were cooled to ambient temperature. Then the beef samples, other main auxiliary ingredients and the soup were sorted and weighed. About 200 g of the mixed samples were vacuum-packed in heat-resistant polyamide-polyethylene bags and then stored at −18 °C for 30 d before reheating. Before reheating, the beef samples were thawed in a refrigerator (4 °C).

#### 2.2.2. Reheating Process

The samples of beef with tangerine peel were divided into four groups for different reheated methods. One group was the control without reheating treatment (CT). The second group was the stir-frying reheating (SF) treatment, in which the samples were subjected to stir-frying in a wok heated by an electromagnetic oven (C22-RT22E01, Midea Group Co., Ltd., Shenzhen, China) with a power of 2200 W for 2.5 min (with stirring at 10 s intervals). The third group was the steaming reheating (SR) treatment, in which the samples were placed on a steamer for steaming for 10 min with boiling water. The fourth group was the microwave reheating (MR) treatment, in which the samples were reheated using a microwave oven (NN-GF351H, Panasonic Appliances Microwave Oven Co., Ltd., Shanghai, China) with a power of 750 W for 2 min. During the reheating process, the final center temperature of the samples was monitored using calibrated plug-in temperature probes (Testo 106, Testo, Black Forest, Germany), and the temperature reached 79.25 ± 1.19 °C. Each experimental group was prepared in triplicate. In this prepared dish, only beef was edible. The rest were merely auxiliary ingredients and not for consumption. Thus, the reheated beef samples were used for further analysis.

### 2.3. Moisture Content and Thiobarbituric Acid Reactive Substance Value

The moisture content of reheated beef samples was determined according to the Chinese national standards [16]. Thiobarbituric acid reactive substance (TBARS) value was measured based on the method of Zhu et al. [17] with slight modifications. The minced sample (10.0 g) was mixed with 50 mL of trichloroacetic acid solution (7.5%, *v*/*v*) and stirred for 30 min. The mixture was subjected to centrifugation at 4000× *g* (4 °C, 20 min) and filtration with filter papers. Then, the filtrate (5.0 mL) was mixed with 5 mL of thiobarbituric acid (0.02 mol/L), and the solution was incubated in a water bath (90 °C, 40 min). The mixture was cooled in an ice bath, and 5 mL of chloroform was added with shaking. The supernatant was taken to measure the absorbance at 532 and 600 nm using a UV–vis spectrophotometer (N4, Shanghai Yi Electrical Analysis Instrument Co., Ltd., Shanghai, China). Three replicate experiments were performed, and the data were averaged.

### 2.4. Gas Chromatography–Mass Spectrometry Analysis

Volatile compounds in different reheated beef samples were analyzed by a gas chromatography–mass spectrometry (GC–MS) instrument (Trace ISQ, Thermofisher, Waltham, MA, USA) equipped with a DB-5MS capillary column (30 m × 0.25 mm × 0.25 µm). The ground beef samples (3 g) were weighed and placed into a 20 mL glass vial, adding 4 mL of saturated NaCl solution and 25 μL of 1,2-dichlorobenzene (internal standard, 1 mg/100 mL). The DVB/CAR/PDMS fiber (50/30 µm thickness, Supelco, Bellefonte, PA, USA) was exposed to the vial at 60 °C for 30 min to absorb the volatile compounds and was desorbed in the GC–MS injection port (250 °C, 7 min). The column temperature program was as follows: the temperature was firstly set at 40 °C for 5 min, heated up to 200 °C at 4 °C/min, and maintained for 2 min, and raised to 250 °C at 20 °C/min and held for 7 min. The MS spectra were performed with ionization energy of 70 eV and a scan mode of 35 to 350 *m*/*z* [18]. The identification of the volatiles was based on the comparison of mass spectra with those in the mass spectra libraries (NIST 11, WILEY 07) and retention indices (RIs) calculated by the n-alkanes (C_7_–C_40_) with the literatures. The concentration of the volatiles was calculated by comparing the peak area with the internal standard.

### 2.5. Odor Activity Value Analysis

In order to assess the contribution of each compound to the overall aroma profile, the odor activity value (OAV) of each substance was determined by dividing its concentration detected in the reheated beef samples by the corresponding odor threshold obtained from the literature [19].

### 2.6. Gas Chromatography–Ion Mobility Spectrometry Analysis

The volatile organic compounds of different reheated beef samples were performed and analyzed using a gas chromatography–ion mobility spectrometry (GC–IMS) system (FlavourSpec^®^, G.A.S., Dortmund, Germany) equipped with a MXT-5 capillary column (15 m × 0.53 mm, 1.0 µm thickness). The ground beef samples (3 g) were transferred into a 20 mL headspace bottle and then incubated at 60 °C for 30 min. Then, the headspace samples (600 µL) were injected into the heated injection port by the automatic sampling system (injection needle temperature, 85 °C). Nitrogen of 99.99% purity was employed as a carrier gas, and its programmed flow was shown as follows: 2 mL/min for 2 min, 10 mL/min for 8 min, 100 mL/min for 10 min and 150 mL/min for 10 min. The total detection time was 30 min. The RIs of the volatile organic compounds were obtained by N-ketones C_4_–C_9_ as external reference.

### 2.7. Determination of Free Amino Acids

The free amino acids (FAAs) in different reheated beef samples were determined according to the method of Zhan et al. [20] with slight modifications. Briefly, 1.0 g of the minced beef sample was mixed with 5% trichloroacetic acid to make up to a 25 mL solution. The solution was sonicated at ambient temperature for 20 min and stood for 2 h. The mixture was filtered using double filter paper, centrifuged at 10,000× *g* for 10 min and filtered by a 0.22 μm membrane. Then, the supernatant (400 μL) was analyzed by an automatic amino acid analyzer (Agilent 1100, Agilent Technologies, Santa Clara, CA, USA).

### 2.8. Determination of Nucleotides

The nucleotides in reheated beef samples were determined by high-performance liquid chromatography (HPLC) based on the method of Jiang et al. [21]. The chopped reheated beef sample (5.0 g) and perchloric acid solution (20 mL, 5% *w*/*w*) were homogenized and extracted by ultrasonic treatment in an ice bath for 5 min. The mixture was centrifuged for 10 min (10,000× *g*, 4 °C), and the supernatant was filtered through a filter paper twice. The pH value of the supernatant was adjusted to 6.8 by adding a 6 mol/L NaOH solution. The solution was diluted to 25 mL with distilled water, filtered by a 0.22 μm membrane and subsequently analyzed using a HPLC system.

### 2.9. Sensory Analysis

Descriptive sensory analysis was used to evaluate the quality of reheated beef samples. The sensory evaluation panelists consisted of 10 trained experts (5 males and 5 females; age 20–30) who had experience in meat sensory research. Prior to sensory evaluation, the panelists were trained four times to be familiar to the samples. Sensory attributes contained flavor, taste, color and texture, and the sensory scoring standard was listed in Table 1. The scale of each attribute ranged from 1 (weakly detectable) to 10 (strongly detectable). During the testing sessions, the reheated beef samples were placed in transparent cups that were marked with three-digit numbers, and purified drink water was applied to clean the palate between samples. The evaluator needed to obtain each sensory score, and each sample was analyzed for three times.

### 2.10. Statistical Analysis

All experiments were conducted in triplicate (*n* = 3). The results were expressed as mean value and standard deviations. Analysis of variance (ANOVA) was performed to evaluate the differences in samples at the level of *p* < 0.05 based on Duncan’s test. The variance analysis was conducted using SPSS software (version 19). The heatmap analysis was carried out by Origin 2022 (Origin Lab Corporation, Northampton, MA, USA).

## 3. Results and Discussion

### 3.1. Moisture Content and Lipid Oxidation

Figure 1A showed the moisture content of samples in the control without reheating treatment (CT), stir-frying reheating (SF), steaming reheating (SR) and microwave reheating (MR) groups. Results indicated that the moisture content of the samples was significantly influenced by various reheating methods (*p* < 0.05). Compared to CT, the moisture content of samples generally decreased after reheating processes. Water is the substance that is prone to be lost during heating treatment. The result was consistent with the previous study reported by Liu et al. [22]. The moisture content of reheated beef samples ranged from 54.28% ± 0.87 in SF samples to 52.62% ± 0.06 in SR samples. The higher moisture content in SF samples might be attributed to its relatively short reheating time, which could reduce the water loss. The thiobarbituric acid reactive substance (TBARS) value was a prominent indicator to measure the lipid oxidation of meat products [23]. Results revealed that there were significant differences in the TBARS values among different reheated beef samples (*p* < 0.05). As presented in Figure 1B, compared to CT samples, reheated beef samples exhibited an increase in TBARS values. This result could be attributed to repeated heating, which accelerated lipid oxidation [7]. SF samples had the highest TBARS value (1.94 mg MDA/kg). Similar result was observed in the study of Li et al. [8]. The reason might be that the high temperature of the stir-frying reheating process accelerated lipid oxidation in beef samples, leading to higher TBARS values [8,24]. For the reheated samples, SR samples presented a lower TBARS value (1.72 mg MDA/kg) compared to other samples. The finding was in accordance with the study of Mei et al., who demonstrated that the TBARS value of braised pork reheated by steaming was lower than that reheated by stir-frying and microwave [23].

### 3.2. Volatile Compounds of Different Reheated Samples by Gas Chromatography–Mass Spectrometry

The volatile compounds of beef samples reheated by different methods were identified by gas chromatography–mass spectrometry (GC–MS), and the results were shown in Table 2 and Figure 2. Fifty-seven volatile compounds were detected in CT and three reheated beef samples, including 17 aldehydes, 23 hydrocarbons, 4 ketones, 4 alcohols, 7 esters and 2 others. Results revealed 53 volatile compounds in CT, 40 compounds in SF samples, 47 compounds in SR samples and 45 compounds in MR samples. The greatest number of volatile compounds was detected in CT samples, and results showed that reheating treatments had an effect on the varieties of volatile compounds in beef samples. As shown in Figure 2, compared to CT samples, the total volatile compound content decreased in reheated beef samples. The total volatile compound content was the highest in CT samples (293.33 μg/kg) and the lowest in MR samples (201.20 μg/kg). Therefore, it could be seen that the types and concentrations of volatile compounds changed in reheated samples when compared to CT samples. The reason might be due to the differences in heat transfer mechanisms and processing conditions in reheating methods [25]. Stir-frying was a dry-heat method, and the heat came from a hot wok. In the steaming reheating process, the samples were heated with the steam generated from boiling water [26]. In the microwave reheating treatment, the heat was generated from the rotation of molecules and ion movements through electromagnetic fields [26,27].

Aldehydes accounted for 71.45% to 89.91% of the total volatile compounds and were the dominant compounds in reheated beef samples. They had an important contribution to the meat aroma due to their low thresholds and high concentrations [28]. There were 17, 16, 12 and 14 aldehydes detected in CT, SF, SR and MR samples, respectively. The total aldehyde content was 221.71 μg/kg in CT samples, and after the reheating process its concentration ranged from 147.11 μg/kg in SR samples to 201.64 μg/kg in SF samples. The total aldehyde concentration in SF samples was higher than that in SR and MR samples. Results demonstrated that the stir-frying reheating treatment was conductive to the retention and production of aldehydes in beef samples. Among the aldehydes, hexanal, octanal and nonanal were the most abundant compounds in CT and reheated beef samples. The result was in accordance with the study of Zhu et al. [17], who found that the three aldehydes were the prominent compounds in air-dried beef at different roasting stages. They imparted the samples with a fatty, fruity, floral and grassy odor [17]. The three aldehydes were mainly generated from oleic acid, linoleic and arachidonic acid degradation [29]. The trend of these three aldehydes was similar to that of the total aldehydes. Results indicated that stir-frying reheating method promoted lipid oxidation to form aldehydes due to its high temperature.

Twenty-three hydrocarbons accounted for 3.20% to 18.31% of the total volatile compounds and were detected in the CT and three reheated samples. Branched hydrocarbons originated from the oxidation of branched fatty acids [30], and terpenes, such as β-pinene, o-cymene, or γ-terpinene, were produced by the biodegradation of substances in tangerine peel or other spices [31,32]. Compared to the CT, reheating treatments significantly decreased the hydrocarbon content. This might be explained by the reheating process that led to the volatilization of hydrocarbons. CT samples had the highest content of hydrocarbons, which was then followed by SR samples. Though hydrocarbons were detected in samples, their contribution to the overall aroma is negligible due to their high odor thresholds [33].

A total of four ketones were detected in the four samples, constituting 1.34% to 3.06% of the total volatile compounds. Ketones were mainly derived from the oxidation of unsaturated fatty acids or the decomposition of amino acids [34]. Ketones had a lesser influence on meat flavor due to their high thresholds [35]. Compared to CT samples, the total ketone content in SR and MR samples increased. SF samples had the lowest concentration of ketones, indicating that stir-frying process led to the loss and degradation of ketones. Two types of ketones were detected in all the samples, including 6-methyl-5-hepten-2-one and α-piperitone. 1-Octen-3-one was only found in CT and SF samples, and 2-undecanone was detected in CT, SR and MR samples.

A total of four alcohols accounted for 2.07% to 8.30% of the total volatile compounds, and they were detected in CT samples and three reheated beef samples. Compared to CT samples, the total alcohol content increased in reheated samples, and the highest concentration was detected in SR samples. 1-Octen-3-ol, formed by lipid oxidation, was an important contributor to the aroma of meat due to its low odor threshold and distinct mushroom odor [36]. In our study, 1-octen-3-ol was only detected in reheated beef samples, and similar result was obtained in reheated braised beef [8]. Terpinen-4-ol and α-terpineol were the predominant alcohols detected in all the samples. It was reported that terpinen-4-ol and α-terpineol were derived from the added spices [37]. The result demonstrated that steaming reheating treatment enhanced the release of terpinen-4-ol and α-terpineol from the spices, and the result was consistent with the reported study of Li et al. [8].

A total of seven esters accounted for 1.57% to 5.68% of the total volatile compounds, and they were identified in four samples. Esters in meat products originated from the esterification of alcohols and carboxylic acids [38]. Compared with CT samples, the ester content increased in SR and MR samples, while decreased in SF samples. The total ester content was the highest in SR samples. Two other compounds were detected in CT and reheated samples, including 3-methyl-2-(2-methyl-2-butenyl)-furan and 2-ethyl-3,5-dimethyl-pyrazine. Among them, 3-methyl-2-(2-methyl-2-butenyl)-furan was found in four samples, and 2-ethyl-3,5-dimethyl-pyrazine was only detected in SF samples. The pyrazines were mainly derived from Strecker degradation at a high temperature [28]. The result revealed that the heating temperature during stir-frying reheating process was relatively high.

### 3.3. Odor-Active Compounds of Reheated Beef Samples

To evaluate the odor-active compounds in reheated beef samples, the odor activity values (OAVs) were calculated, and the result was listed in Table 3. The volatile compounds with OAVs greater than one were considered as the primary contributors to the aroma of samples, and the higher the OAV, the greater the influence on the aroma profile [34]. Results showed that fifteen odor-active compounds with OAV > 1 were identified in CT and reheated beef samples, including 11 aldehydes, 1 hydrocarbon, 1 ketone, 1 alcohol and 1 pyrazine. These compounds significantly contributed to the overall aroma of reheated beef samples. There were 12, 14, 8 and 8 odor-active compounds identified in CT, SF, SR and MR samples, respectively. Aldehydes, which mainly had fatty, fruity, floral and oily odor, were the most prominent contributors to the aroma. Seven aldehydes were detected in CT and three reheated samples, including hexanal, heptanal, octanal, nonanal, (E)-2-nonenal, decanal and (E)-2-decenal. The result aligned with the previous study that the seven aldehydes were also the main odor-active compounds of precooked stewed beef and reheated beef [28]. The OAVs of nonanal (fatty and floral) and octanal (citrus and fatty) were high in all the samples. The OAVs of nonanal and octanal in SR and MR samples were lower than those in the other samples. This might be the reason why SR and MR samples exhibited lower flavor scores during sensory evaluation. The OAVs of the majority of odor-active compounds in SR samples were lower than those in CT samples, suggesting that its aroma profile differed from that of CT samples, and the steaming reheating process led to the degradation or loss of odor-active compounds due to its long reheating time. In MR samples, only three odor-active compounds had OAVs higher than those in CT samples. In comparison to CT samples, SF samples had higher OAVs of benzeneacetaldehyde, (E)-2-decenal, 2,4-decadienal, (E,E)-2,4-decadienal, 1-octen-3-one, 1-octen-3-ol and 2-ethyl-3,5-dimethyl-pyrazine. Furthermore, among the reheated samples, odor-active compounds, such as 2,4-decadienal, (E,E)-2,4-decadienal, 1-octen-3-one and 2-ethyl-3,5-dimethyl-pyrazine, were only detected in SF samples. This might be the key factor that caused the differences in aroma profile between SF samples and other samples. Benzeneacetaldehyde, which had a sweet and floral aroma, was generated from the degradation of L-phenylalanine [39]. 2-Ethyl-3,5-dimethyl-pyrazine derived from Strecker degradation had a roasted and caramel aroma [40]. The other odor-active compounds had been reported to be the main odorants of reheated precooked stewed beef [28]. Based on the above analysis, SF samples contained the most varieties of odor-active compounds, and the OAVs of most of these odor-active compounds were relatively higher among the reheated samples.

To further explore the differences among beef samples, a heatmap with cluster analysis was conducted according to the OAVs of odor-active compounds. As shown in Figure 3, SR and MR samples were grouped together, indicating that the differences in odor-active compounds between these two samples were relatively minor. CT and SF samples were clustered into a single group, suggesting that stir-frying reheating treatment could maintain the original flavor characteristics of beef samples. Based on the heatmap analysis, the odor-active compounds could be clustered into four groups. Cluster 1 contained hexanal and heptanal. The difference in MR samples might be due to the higher OAVs of these two components. Cluster 2 consisted of octanal, nonanal, dodecanal, (E)-2-decenal, 2,4-decadienal, 1-octen-3-one, benzeneacetaldehyde, (E)-2-nonenal and decanal. The majority of these odor-active compounds demonstrated higher OAVs in CT and SF samples. This might be the reason for the flavor differences between these two samples and the other samples. Cluster 3 was composed of (E,E)-2,4-decadienal, 2-ethyl-3,5-dimethyl-pyrazine and 1-octen-3-ol. The OAVs of these odor-active compounds were the highest in SF samples, which might account for the distinct flavor profiles that differentiated the SF samples from the other samples. Cluster 4 included β-pinene, which exhibited the highest OAVs in CT samples.

### 3.4. Changes of Volatile Organic Compounds by Gas Chromatography–Ion Mobility Spectrometry

Gas chromatography–ion mobility spectrometry (GC–IMS) analysis was conducted to investigate the flavor differences in the three beef samples prepared using different reheating treatments. The topographic plots obtained from GC–IMS in the three reheated beef samples were shown in Figure 4. From Figure 4A, the spots in the fingerprint exhibited the volatile organic compounds, and the spot colors showed their concentration (red meant a peak of high intensity and white meant a peak of relatively low intensity). The *x*-axis and *y*-axis denoted the ion migration time and retention time of GC, respectively. The majority of the volatile organic compounds in different reheated beef samples were detected in the drift time of 6.0–9.0 ms and the retention time of 200–1400 s. Result indicated that the types of volatile organic compounds in different reheated samples were similar;however their concentrations varied.

To clearly distinguish the differences among reheated beef samples, SF samples were taken as the reference, and the spectra of others (SR and MR samples) were deducted from SF samples (Figure 4B). Different points represented different volatile organic compounds. If the content of volatile organic compounds was the same among samples, the background was white, red spots indicated that the concentration of the compounds was higher than that in the reference (SF samples), and blue spots meant the content of the compounds was lower than that in SF samples [45]. Compared with SF samples, the majority of the volatile organic compounds exhibited higher signal intensities in SR samples in the retention time of 600–800 s and 1200–1400 s, while most volatile organic compounds had decreased signal intensities in MR samples in the range of 800–1200 s, indicating that reheating treatments had an effect on the production of volatile organic compounds.

To directly compare the changes in the content of volatile organic compounds, the fingerprint spectrum was established, and the result was shown in Figure 4C. Each row meant a signal peak, and each column meant a component. The change in the concentration of each component was reflected by the color change at each point, in which the lighter the color, the lower the concentration. As shown in Figure 4C, a total of 52 volatile organic compounds (include their monomers and dimers) were identified by GC–IMS analysis, including 15 aldehydes, 14 alcohols, 12 ketones, 3 esters, 3 hydrocarbons, 2 acids and 2 furans and 1 nitrogen-containing compound. The fingerprint spectrum was divided into four regions, labeled a (purple), b (green), c (yellow) and d (red). In region a, fourteen volatile organic compounds exhibited high concentration in the three reheated beef samples, predominantly composed of alcohols, aldehydes and ketones. In region b, the compounds, including (E)-2-heptenal, pentanal-D, pentanal-M, 1-octen-3-one, morpholine, ethyl 2-hydroxypropanoate, butanal-D, acetic acid-M, acetic acid-D, 1-octen-3-ol, 1-penten-3-ol, heptaldehyde-M and 1-hexanal-D, had the highest concentration in SF samples. Among them, most of the aldehydes and alcohols were aroma-active compounds in cooked and reheated stewed beef [28]. These compounds might be correlated with the better flavor of reheated beef samples. In region c, the relative concentrations of heptaldehyde-D, 3-methyl-2-butenal, 1-nonanal-D and ethyl heptanoate increased after the microwave reheating process. Ping et al. [9] found that microwave reheating could elevate the concentration of 3-methyl-2-butenal in ceramic-pot sealed meat. In region d, SR samples contained higher levels of beta-pinene, beta-myrcene and (+)-limonene and numerous alcohols and ketones. Beta-pinene presented a rosin and resin aroma [9], and beta-myrcene exhibited an aroma reminiscent of herbs and pungency [32]. These three terpenes originated from the added spices [31,32]. Given their cyclic structures, terpenes exhibited heat sensitivity and were easily degraded or rearranged under high temperature heating conditions [46]. The result demonstrated that steaming reheating treatments effectively preserved the volatile organic compounds of the spices, which was consistent with the findings from GC–MS analysis.

To further highlight the olfactory variations in different reheated beef samples, PCA analysis was carried out based on the peak intensities of volatile organic compounds by GC–IMS. As shown in Figure 5, the cumulative variance contribution of the two principal components was 90%, and PC1 and PC2 accounted for 66% and 24% of the total variance, respectively. SR samples were located on the negative PC1 and PC2 position, MR samples were located on the positive PC1 and PC2 axis, and SF samples were located on the positive PC1 axis and negative PC2 axis. Therefore, results demonstrated that the three reheated beef samples were clearly separated in the distribution map, suggesting that the aroma characteristics of beef samples reheated by stir-frying, steaming and microwave treatments considerably differed.

### 3.5. Free Amino Acid Concentration in Reheated Beef Samples

Free amino acids (FAAs) had an important contribution to the taste properties in meat products due to their own taste and synergistic interaction with other flavor components [47]. The composition and content of FAAs in beef samples subjected to different reheating methods were shown in Table 4. A total of seventeen FAAs were detected. All of them were detected in reheated beef samples, and fifteen FAAs were present in CT samples. Compared to CT, the reheating treatments resulted in a significant increase in the content of FAAs. The content of each FAA increased in reheated beef samples except arginine (Arg) and methionine (Met). The total FAA concentration was the highest in SF samples, but there were no significant difference among SF and MR samples (*p* > 0.05). The total FAA content in these two samples was significantly higher than that in SR and CT samples. The result revealed that the stir-frying reheating process facilitated protein degradation due to its high heating temperature, thereby promoting the release of FAAs in beef samples. Among the amino acids, glutamic acid (Glu) and aspartic acid (Asp) presented an umami taste. The content of Glu and Asp was significantly higher in MR samples than that in other samples. The result was consistent with the previous study of Ping et al. [48], who found that the microwave reheating process promoted the release of Glu and Asp in Yu-Shiang Shredded pork compared to other reheating methods (water bath reheating, stir-frying reheating and steam reheating). It was reported that microwave heating could lead to rapid molecular excitation and protein degradation to enhance Glu release [49]. The amino acids serine (Ser), glycine (Gly), threonine (Thr), alanine (Ala) and proline (Pro) had a sweet taste, while histidine (His), Arg, tyrosine (Tyr), valine (Val), Met, phenylalanine (Phe), isoleucine (Ile), leucine (Leu) and lysine (Lys) exhibited a bitter taste [50]. The highest content of both sweet and bitter amino acids were detected in SF samples.

### 3.6. Nucleotides Concentration in Reheated Beef Samples

The composition and content of 5′-nucleotides in beef samples prepared by different reheating methods were listed in Table 5. Five types of 5′-nucleotides were detected in the three reheated beef samples, namely 5′-CMP, 5′-AMP, 5′-UMP, 5′-GMP and 5′-IMP, while four 5′-nucleotides were identified in CT samples. Results demonstrated that reheating treatments had significant differences on the 5′-nucleotide content in beef samples. CT samples showed the lowest 5′-nucleotide content (154.49 mg/100 g). SF samples had the highest 5′-nucleotide concentration (214.29 mg/100 g), followed by MR samples (168.59 mg/100 g). 5′-AMP and 5′-IMP were important 5′-nucleotides that could provide umami taste to meat products [7]. Moreover, 5′-AMP, 5′-IMP and 5′-GMP had a synergistic effect to promote umami taste [51]. Results showed that the content of 5′-AMP, 5′-IMP and 5′-GMP was the highest after the stir-frying reheating process, suggesting that this reheating treatment promoted the ATP degradation in beef samples.

### 3.7. Sensory Evaluation of Different Reheated Beef Samples

The sensory results of beef samples prepared by different reheating methods were listed in Table 6. The sensory attributes of reheated beef samples contained flavor, taste, color and texture. The reheating treatments had a significant effect on the sensory scores of all attributes. For the reheated samples, all sensory indicators demonstrated a decline compared to those of CT samples. The result was in accordance with the previous study of Li et al. [8]. The flavor score of SF samples (8.11) was significantly higher than that of MR samples (7.08) and SR samples (6.21). The result was consistent with those of previously identified odor-active compounds. Moreover, SF samples had the highest taste score compared to other reheated samples. Among the reheated samples, MR samples had the highest color and texture scores, which might be due to its shorter reheating time. As shown in Table 6, it could be seen that SF samples had the highest flavor and taste scores, while MR samples exhibited the highest scores of color and texture attributes among the reheated samples.

## 4. Conclusions

The present study investigated the effects of different reheating methods (stir-frying reheating, steaming reheating and microwave reheating) on the edible quality and flavor characteristics of the prepared beef with tangerine peel dish. The three different reheating treatments had an important influence on the quality of prepared beef with tangerine peel. For the reheated samples, the stir-frying reheating samples had a higher moisture content and the highest TBARS value due to its short reheating time and high temperature. The stir-frying reheating samples had better aroma characteristics because they contained more varieties of odor-active compounds, and the OAVs of most of these compounds were relatively high. Except that, the stir-frying reheating samples exhibited the highest concentrations of both free amino acids and 5’-nucleotides. Among the reheated samples, the stir-frying reheating samples received the highest scores of flavor and taste attributes during the sensory evaluation. In conclusion, stir-frying was the best reheating method to obtain superior aroma and taste quality for the prepared beef with tangerine peel dish.

Beef was the third most consumed meat globally due to its rich nutritional ingredients, such as protein, amino acids, vitamins and minerals. Beef with tangerine peel is a well-known traditional dish. This dish appeared in the market as a prepared product. Reheating methods had an effect on the quality of prepared dishes. Our study aimed to evaluate the flavor changes of the prepared beef with tangerine peel dish subjected to different reheating treatments. The findings could provide consumers, retailers and caterers with valuable information and insights on reheating the prepared beef with tangerine peel dish and other prepared meat products. Future research could focus on elucidating the mechanisms of flavor formation and assessing the nutritional properties in the prepared beef with tangerine peel dish using different reheating methods.

## Figures and Tables

**Figure 1 foods-14-03017-f001:**
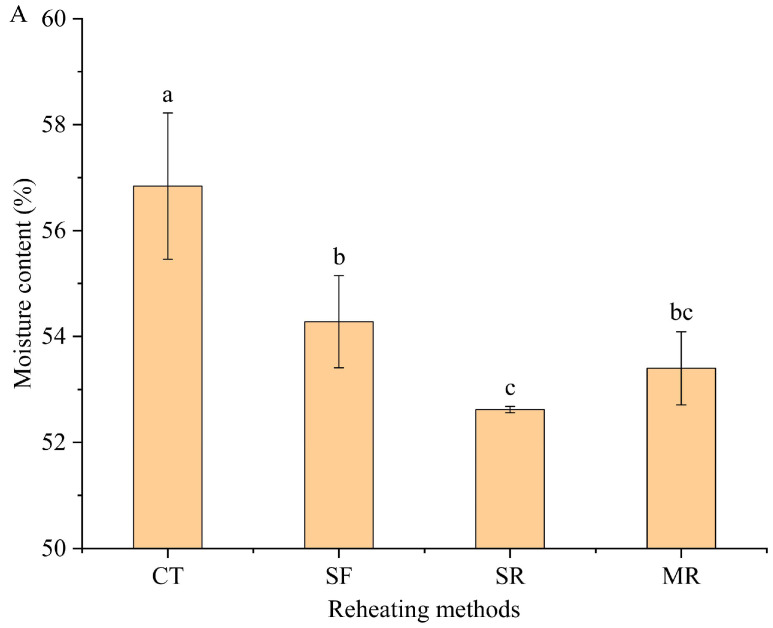
Effect of different reheating methods on moisture content (**A**) and TBARS value (**B**) in beef samples. Each experimental group was repeated three times. Different lowercase letters (a–c) represented a significant difference (*p* < 0.05). CT, control without reheating treatment; SF, stir-frying reheating; SR, steaming reheating; MR, microwave reheating.

**Figure 2 foods-14-03017-f002:**
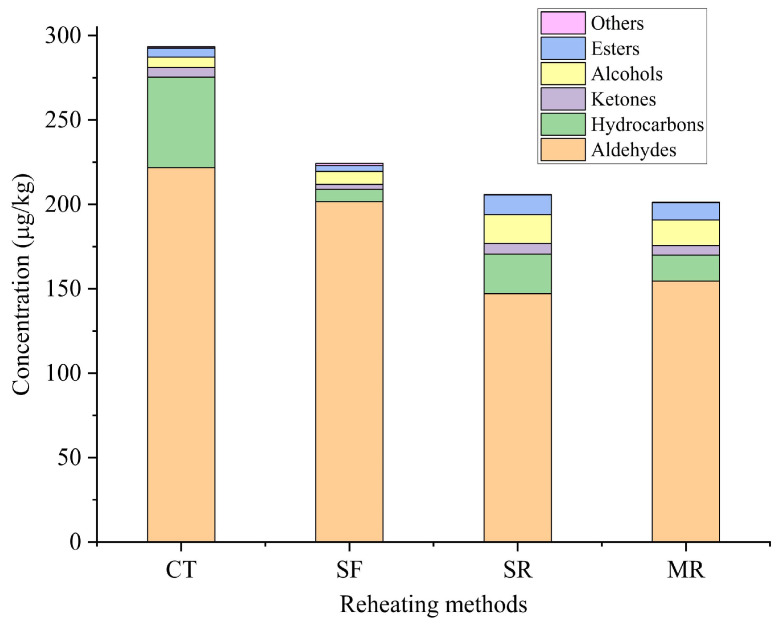
The concentration of each chemical of the volatile compounds in the beef samples reheated by different methods. CT, control without reheating treatment; SF, stir-frying reheating; SR, steaming reheating; MR, microwave reheating.

**Figure 3 foods-14-03017-f003:**
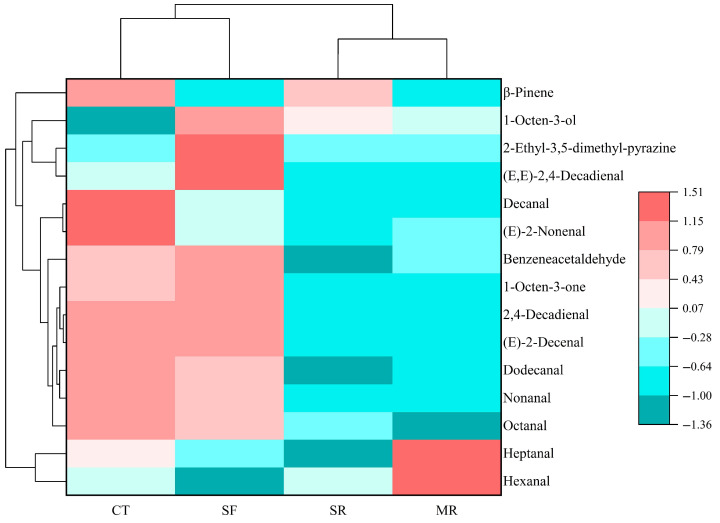
Heatmap analysis of the odor-active compounds in beef samples prepared by different reheating methods. CT, control without reheating treatment; SF, stir-frying reheating; SR, steaming reheating; MR, microwave reheating.

**Figure 4 foods-14-03017-f004:**
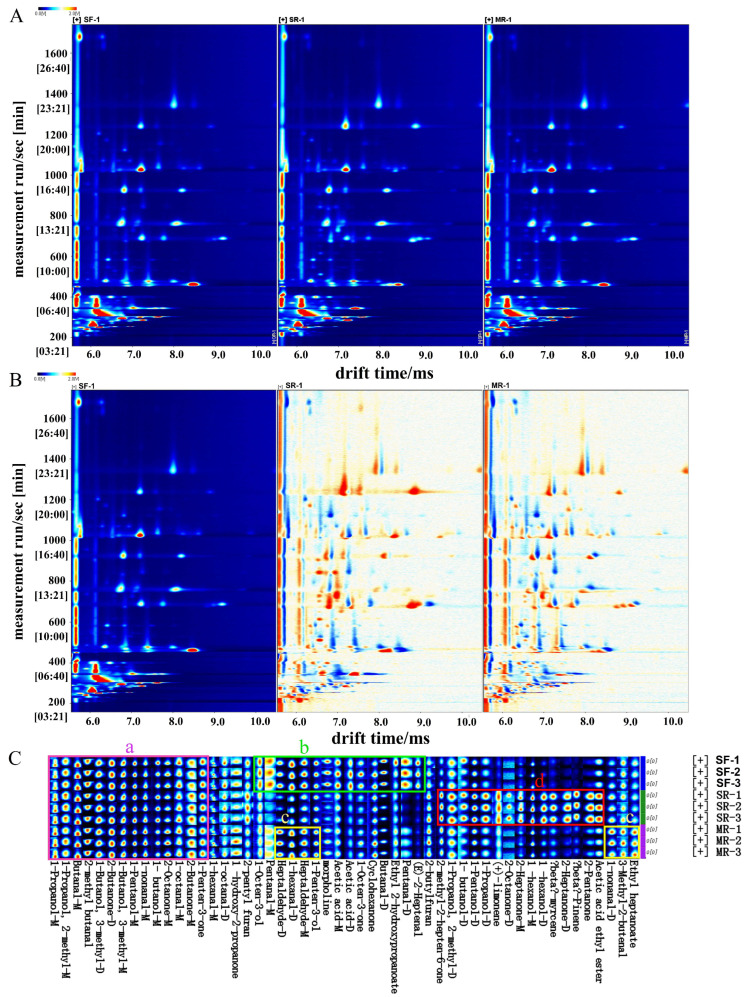
GC–IMS spectra of beef samples prepared by different reheating methods. (**A**) Two dimensional topographic plot; (**B**) comparison of differences (SF samples as reference); (**C**) fingerprint of volatile organic compounds. Each experimental group was repeated three times. SF, stir-frying reheating; SR, steaming reheating; MR, microwave reheating.

**Figure 5 foods-14-03017-f005:**
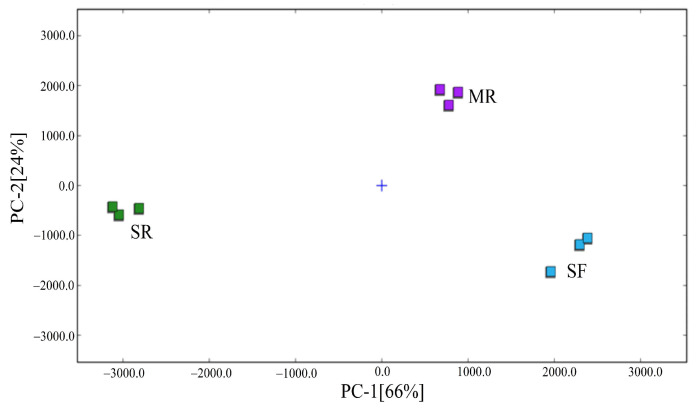
PCA analysis obtained from the peak intensities of volatile organic compounds detected by GC–IMS. SF, stir-frying reheating; SR, steaming reheating; MR, microwave reheating.

**Table 1 foods-14-03017-t001:** Sensory evaluation criteria.

Sensory Attribute	Standard Description	Scores
Flavor	Obvious meaty and fatty flavor, no peculiar smell	7–10
	Normal meaty and fatty flavor, no peculiar smell	4–6
	Less meaty and fatty flavor, peculiar smell	1–3
Taste	Rich meaty taste, full aftertaste, no bitter and sour taste	7–10
	Moderate meaty taste, no bitter and sour taste	4–6
	Mild or no meaty taste, bitter and sour taste	1–3
Color	Bright color, uniform appearance with gloss	7–10
	Relative bright color, slightly uneven appearance with gloss	4–6
	Dark color, uneven appearance without gloss	1–3
Texture	Complete meat, close texture, moderate hardness	7–10
	Relative complete meat, close texture, softness or hardness	4–6
	Incomplete meat, loose texture, particular softness or hardness	1–3

**Table 2 foods-14-03017-t002:** Volatile compounds of beef samples reheated by different methods.

Code	Compounds	RI ^1^	RI ^2^	Concentration (µg/kg) ^3^			
				CT	SF	SR	MR
	Aldehydes (17)						
1	Hexanal	802	829	36.77 ± 0.50 a	27.57 ± 0.51 a	36.12 ± 18.50 a	44.94 ± 8.51 a
2	Heptanal	908	927	10.28 ± 0.57 ab	9.03 ± 1.32 ab	6.86 ± 1.49 b	13.33 ± 4.26 a
3	Octanal	1001	995	29.39 ± 3.92 a	26.08 ± 7.66 ab	19.99 ± 2.73 bc	16.86 ± 1.54 c
4	Benzeneacetaldehyde	1044	1037	4.13 ± 0.13 b	5.74 ± 0.85 a	nd ^4^	2.04 ± 0.30 c
5	Nonanal	1102	1085	123.52 ± 21.45 a	113.51 ± 21.43 a	65.81 ± 13.32 b	62.51 ± 8.64 b
6	Citronellal	1153	1131	0.28 ± 0.04 a	nd ^4^	nd ^4^	nd ^4^
7	(E)-2-Nonenal	1166	1135	1.61 ± 0.13 a	0.79 ± 0.19 b	0.48 ± 0.13 c	0.59 ± 0.19 bc
8	Decanal	1205	1181	5.02 ± 0.95 a	3.31 ± 0.95 bc	2.60 ± 0.14 c	2.63 ± 0.14 c
9	4-(1-Methylethyl)-benzaldehyde	1240	1208	0.25 ± 0.02 a	0.13 ± <0.01 b	0.26 ± 0.05 a	0.26 ± 0.02 a
10	β-Citral	1238	1211	1.69 ± 0.29 c	2.82 ± 0.12 b	5.73 ± 1.51 a	3.79 ± 0.19 ab
11	(E)-2-Decenal	1263	1230	2.19 ± 0.69 a	2.21 ± 0.21 a	0.72 ± 0.26 b	0.87 ± 0.16 b
12	α-Citral	1270	1238	1.22 ± 0.15 c	4.19 ± 0.19 b	7.94 ± 2.13 a	5.46 ± 0.35 ab
13	2,4-Decadienal	1309	1277	0.71 ± 0.07 a	0.76 ± 0.11 a	nd ^4^	nd ^4^
14	2-Undecenal	1368	1329	2.46 ± 0.42 a	2.02 ± 0.27 b	nd ^4^	0.54 ± 0.14 c
15	Undecanal	1310	1279	0.95 ± 0.12 a	0.65 ± 0.31 b	0.38 ± 0.07 b	0.42 ± 0.04 b
16	(E,E)-2,4-Decadienal	1313	1277	0.63 ± <0.01 b	2.30 ± 0.09 a	nd ^4^	nd ^4^
17	Dodecanal	1409	1378	0.61 ± 0.02 a	0.53 ± 0.53 b	0.22 ± 0.04 c	0.28 ± 0.01 c
	subtotal			221.71	201.64	147.11	154.52
	Hydrocarbons (23)						
18	p-Xylene	872	849	0.53 ± 0.08 d	1.16 ± 0.07 a	0.74 ± 0.06 c	0.87 ± 0.02 b
19	Ethylbenzene	878	841	1.14 ± 0.06 b	nd ^4^	0.50 ± 0.19 c	3.44 ± 1.44 a
20	o-Xylene	890	854	0.12 ± 0.01 c	0.41 ± 0.01 a	0.06 ± 0.04 c	0.22 ± 0.11 b
21	α-Pinene	939	914	0.38 ± 0.08 b	nd ^4^	0.61 ± 0.15 a	0.20 ± <0.01 c
22	Camphene	955	928	0.78 ± 0.10 b	nd ^4^	1.10 ± 0.31 a	0.50 ± 0.02 c
23	β-Pinene	980	973	4.17 ± 0.19 a	nd ^4^	2.93 ± 0.17 b	nd ^4^
24	o-Cymene	1022	1005	0.52 ± 0.07 c	nd ^4^	1.05 ± 0.07 a	0.77 ± 0.03 b
25	p-Cymene	1025	1006	0.22 ± 0.03 c	0.47 ± 0.13 b	nd ^4^	0.62 ± <0.01 a
26	β-Cymene	1026	1006	0.52 ± 0.03 a	nd ^4^	0.33 ± <0.01 b	nd ^4^
27	Sylvestrene	1027	1007	0.79 ± 0.02 b	0.31 ± 0.10 c	1.42 ± 0.07 a	0.40 ± 0.24 c
28	2-Methyl-decane	1061	1045	0.73 ± 0.03 c	0.70 ± 0.08 c	1.31 ± 0.10 b	3.00 ± 1.81 a
29	3-Methyl-decane	1069	1052	2.38 ± 0.10 a	0.84 ± 0.03 d	1.30 ± 0.02 b	1.17 ± 0.01 c
30	(E)-β-Ocimene	1036	1022	9.25 ± 1.61 a	nd ^4^	1.24 ± 0.33 b	nd ^4^
31	(Z)-β-Ocimene	1037	1030	4.15 ± 0.15 a	nd ^4^	1.22 ± 0.12 b	0.46 ± <0.01 c
32	4-Methyl-decane	1059	1040	1.16 ± 0.17 a	nd ^4^	0.44 ± 0.06 b	0.39 ± 0.02 b
33	γ-Terpinene	1062	1040	nd ^4^	nd ^4^	3.27 ± 0.24 a	nd ^4^
34	α-Terpinolene	1088	1069	1.38 ± 0.28 a	nd ^4^	1.38 ± 0.24 a	1.14 ± 0.20 a
35	(E,E)-alloocimene	1140	1110	5.22 ± 1.33 a	nd ^4^	1.42 ± 0.26 b	0.75 ± 0.05 c
36	3-Methyl-undecane	1169	1149	0.63 ± 0.09 c	0.82 ± 0.14 bc	1.40 ± 0.11 a	1.07 ± 0.37 ab
37	2,4-Dimethyl-undecane	1213	1185	14.61 ± 2.09 a	nd ^4^	nd ^4^	nd ^4^
38	2,6-Dimethyl-undecane	1210	1189	3.66 ± 0.15 a	1.13 ± 0.20 b	1.40 ± 0.18 b	nd ^4^
39	3-Methyl-tridecane	1375	1336	0.47 ± 0.04 a	0.41 ± 0.12 ab	0.31 ± 0.03 b	0.37 ± 0.06 ab
40	2,6,11-Trimethyl-dodecane	1275	1294	0.90 ± 0.10 a	0.92 ± 0.06 a	nd ^4^	nd ^4^
	subtotal			53.71	7.17	23.43	15.37
	Ketones (4)						
41	1-Octen-3-one	980	970	0.40 ± 0.10 b	0.53 ± 0.03 a	nd ^4^	nd ^4^
42	6-Methyl-5-hepten-2-one	986	980	3.14 ± 0.14 a	1.35 ± 0.24 c	2.14 ± 0.28 b	1.41 ± 0.28 c
43	α-Piperitone	1252	1222	1.53 ± 0.43 b	1.12 ± 0.32 b	3.64 ± 0.85 a	3.92 ± 0.38 a
44	2-Undecanone	1287	1267	0.62 ± 0.08 a	nd ^4^	0.52 ± 0.14 ab	0.40 ± 0.06 b
	subtotal			5.69	3.00	6.30	5.73
	Alcohols (4)						
45	1-Octen-3-ol	982	973	nd ^4^	3.65 ± 0.25 a	2.53 ± 0.42 b	2.12 ± 0.25 b
46	Terpinen-4-ol	1177	1153	2.43 ± 0.05 b	1.59 ± 0.30 c	5.54 ± 1.33 a	4.81 ± 0.44 a
47	α-Terpineol	1189	1166	3.65 ± 0.31 b	2.45 ± 0.47 c	8.35 ± 1.93 a	7.80 ± 0.48 a
48	Neodihydrocarveol	1220	1170	nd ^4^	nd ^4^	0.67 ± 0.16 a	0.32 ± <0.01 b
	subtotal			6.08	7.69	17.09	15.05
	Esters (7)						
49	Linalyl acetate	1258	1235	0.46 ± 0.02 b	0.45 ± 0.05 b	0.74 ± 0.19 a	0.72 ± 0.09 a
50	Bornyl acetate	1267	1261	0.36 ± 0.02 b	0.24 ± <0.01 c	0.89 ± 0.22 a	0.45 ± 0.08 b
51	Myrtenyl acetate	1325	1300	0.23 ± <0.01 a	nd ^4^	0.24 ± 0.06 a	0.21 ± 0.04 a
52	2-Acetoxy-1,8-cineole	1344	1315	0.84 ± 0.03 b	0.51 ± 0.01 c	1.26 ± 0.27 a	1.28 ± 0.09 a
53	α-Terpinyl acetate	1350	1323	2.42 ± 0.06 b	1.50 ± 0.09 c	5.26 ± 1.22 a	5.10 ± 0.76 a
54	β-Citronellol acetate	1354	1327	0.27 ± <0.01 b	0.19 ± 0.07 b	0.80 ± 0.19 a	0.65 ± 0.11 a
55	Geranyl acetate	1384	1356	0.78 ± 0.03 c	0.63 ± 0.19 c	2.50 ± 0.36 a	1.87 ± 0.27 b
	subtotal			5.36	3.52	11.69	10.28
	Others (2)						
56	3-Methyl-2-(2-methyl-2-butenyl)-furan	1093	1079	0.78 ± 0.07 a	0.26 ± <0.01 b	0.28 ± 0.03 b	0.25 ± 0.10 b
57	2-Ethyl-3,5-dimethyl-pyrazine	1095	1081	nd ^4^	0.99 ± 0.01 a	nd ^4^	nd ^4^
	subtotal			0.78	1.25	0.28	0.25

^1^ RIs (retention indexes) were obtained from the website (https://webbook.nist.gov (accessed on 24 May 2025)). ^2^ RIs were calculated on the DB-5MS column relative to n-alkanes. ^3^ Data were listed as mean ± standard deviation (*n* = 3). Different lowercase letters (a–d) in the same row represented significant differences (*p* < 0.05). CT, control without reheating treatment; SF, stir-frying reheating; SR, steaming reheating; MR, microwave reheating. ^4^ nd, not detected.

**Table 3 foods-14-03017-t003:** Odor-active compounds of beef samples reheated by different methods.

Code ^1^	Compounds	Odor Descriptions	Odor Threshold (μg/kg)	OAV ^2^			
				CT	SF	SR	MR
1	Hexanal	Green, grassy [41]	4.5	8.17	6.13	8.03	9.99
2	Heptanal	Fatty, fruity [41]	3	3.43	3.01	2.29	4.44
3	Octanal	Citrus, fatty [41]	0.7	41.99	37.26	28.56	24.09
4	Benzeneacetaldehyde	Sweet, floral [42]	4	1.03	1.44	nd	<1
5	Nonanal	Fatty, floral [41]	1.1	112.29	103.19	59.83	56.83
7	(E)-2-Nonenal	Fatty, cucumber [41]	0.19	8.47	4.16	2.53	3.11
8	Decanal	Fatty, sweet [25]	0.3	16.73	11.03	8.67	8.77
11	(E)-2-Decenal	Oily [25]	0.3	7.30	7.37	2.40	2.90
13	2,4-Decadienal	Oily, fatty [43]	0.3	2.37	2.53	nd	nd
16	(E,E)-2,4-Decadienal	Fatty, fruity [25]	0.07	9.00	32.86	nd	nd
17	Dodecanal	Fatty, citrus [44]	0.53	1.15	1.00	<1	<1
23	β-Pinene	Rosin, resin [9]	6	<1	nd	<1	nd
41	1-Octen-3-one	Mushroom, metallic [43]	0.05	8.00	10.60	nd	nd
45	1-Octen-3-ol	Mushroom [41]	1	nd	3.65	2.53	2.12
57	2-Ethyl-3,5-dimethyl-pyrazine	Roasted, caramel [40]	0.16	nd	6.19	nd	nd

^1^ The code of odor-active compounds corresponded to that in Table 2. ^2^ CT, control without reheating treatment; SF, stir-frying reheating; SR, steaming reheating; MR, microwave reheating. nd, not detected.

**Table 4 foods-14-03017-t004:** Effect of different reheating treatments on free amino acid content in beef samples.

Free Amino Acid	Taste Characteristics	Concentration (mg/100 mL)			
		CT	SF	SR	MR
Aspartic acid (Asp)	Umami	0.39 ± 0.01 c	2.05 ± 0.09 a	1.84 ± 0.05 b	2.10 ± 0.04 a
Glutamic acid (Glu)	Umami	3.80 ± 0.29 d	11.31 ± 0.10 b	10.02 ± 0.01 c	12.27 ± 0.67 a
Serine (Ser)	Sweet	0.73 ± 0.12 c	1.93 ± 0.02 a	1.58 ± 0.01 b	1.84 ± 0.06 a
Glycine (Gly)	Sweet	0.18 ± 0.03 c	1.26 ± 0.03 a	1.05 ± 0.02 b	1.25 ± 0.02 a
Threonine (Thr)	Sweet	0.34 ± 0.02 c	1.54 ± 0.04 a	1.26 ± 0.05 b	1.52 ± 0.03 a
Alanine (Ala)	Sweet	1.16 ± 0.03 c	3.50 ± 0.26 a	2.92 ± 0.02 b	3.55 ± 0.31 a
Proline (Pro)	Sweet	0.70 ± 0.01 c	2.48 ± 0.19 a	1.64 ± 0.29 b	1.92 ± 0.07 b
Histidine (His)	Bitter	0.27 ± 0.02 b	0.53 ± 0.07 a	0.48 ± 0.04 a	0.55 ± 0.03 a
Arginine (Arg)	Bitter	9.37 ± 0.01 a	1.72 ± 0.18 b	1.28 ± 0.01 d	1.53 ± 0.09 c
Tyrosine (Tyr)	Bitter	nd	1.53 ± 0.11 a	1.28 ± 0.09 b	1.40 ± 0.10 ab
Valine (Val)	Bitter	0.49 ± 0.01 c	2.05 ± 0.20 a	1.58 ± 0.07 b	1.94 ± 0.02 a
Methionine (Met)	Bitter	0.23 ± 0.05 a	0.14 ± 0.03 b	0.11 ± 0.01 b	0.15 ± 0.01 b
Phenylalanine (Phe)	Bitter	0.46 ± 0.01 c	1.82 ± 0.08 ab	1.60 ± 0.10 b	1.93 ± 0.15 a
Isoleucine (Ile)	Bitter	0.47 ± 0.01 c	1.58 ± 0.12 a	1.28 ± 0.14 b	1.58 ± 0.10 a
Leucine (Leu)	Bitter	0.85 ± 0.02 c	2.78 ± 0.20 a	2.34 ± 0.15 b	2.72 ± 0.04 a
Lysine (Lys)	Bitter	0.43 ± 0.07 c	1.85 ± 0.02 a	1.43 ± 0.03 b	1.77 ± 0.02 a
Cysteine (Cys-s)	Tasteless	nd	0.02 ± <0.01 a	0.02 ± 0.01 a	0.03 ± 0.01 a
Umami FAAs		4.18 ± 0.29 d	13.36 ± 0.19 b	11.86 ± 0.05 c	14.37 ± 0.71 a
Sweet FAAs		3.11 ± 0.22 c	10.71 ± 0.51 a	8.45 ± 0.27 b	10.07 ± 0.48 a
Bitter FAAs		12.54 ± 0.07 bc	13.98 ± 0.98 a	11.39 ± 0.55 c	13.57 ± 0.54 ab
Total		19.83 ± 0.56 c	38.09 ± 1.67 a	31.71 ± 0.87 b	38.04 ± 1.72 a

Data were expressed as mean ± standard deviation (*n* = 3). Different lowercase letters (a–d) in the same row showed a significant difference (*p* < 0.05). nd, not detected. CT, control without reheating treatment; SF, stir-frying reheating; SR, steaming reheating; MR, microwave reheating.

**Table 5 foods-14-03017-t005:** Effect of different reheating treatments on 5′-nucleotide content in beef samples.

5′-Nucleotide	Concentration (mg/100 g)			
	CT	SF	SR	MR
5′-CMP	nd	3.96 ± 0.09 a	2.08 ± 0.07 b	1.95 ± 0.16 b
5′-AMP	7.07 ± 0.02 c	13.63 ± 1.28 a	8.10 ± 0.25 b	7.58 ± 0.21 b
5′-UMP	140.69 ± 0.02 b	185.97 ± 9.07 a	142.71 ± 12.61 b	152.57 ± 4.52 b
5′-GMP	2.61 ± 0.06 b	3.13 ± 0.08 a	2.22 ± 0.08 c	2.14 ± 0.13 c
5′-IMP	4.12 ± 0.12 c	7.61 ± 0.02 a	4.90 ± 0.22 b	4.36 ± 0.21 c
Total	154.49 ± 0.03 c	214.29 ± 10.52 a	160.01 ± 13.23 b	168.59 ± 5.21 b

Data were expressed as mean ± standard deviation (*n* = 3). Different lowercase letters (a–c) in the same row represented significant differences (*p* < 0.05). nd, not detected. CT, control without reheating treatment; SF, stir-frying reheating; SR, steaming reheating; MR, microwave reheating.

**Table 6 foods-14-03017-t006:** Effect of different reheating methods on sensory evaluation result of beef samples.

Sensory Attributes	CT	SF	SR	MR
Flavor	8.80 ± 0.34 a	8.11 ± 0.21 b	6.21 ± 0.32 d	7.08 ± 0.18 c
Taste	7.50 ± 0.41 a	7.54 ± 0.30 a	5.74 ± 0.46 c	6.52 ± 0.27 b
Color	7.82 ± 0.25 a	6.87 ± 0.33 c	7.05 ± 0.28 bc	7.13 ± 0.15 b
Texture	7.66 ± 0.48 a	6.63 ± 0.22 b	6.34 ± 0.20 b	7.01 ± 0.44 ab

Data were expressed as mean ± standard deviation (*n* = 3). Different lowercase letters (a–d) in the same row presented a significant difference (*p* < 0.05). CT, control without reheating treatment; SF, stir-frying reheating; SR, steaming reheating; MR, microwave reheating.

## Data Availability

The original contributions presented in the study are included in the article; further inquiries can be directed to the corresponding authors.

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
