# Peer review of "Comparative Analysis of Flavor Quality of Beef with Tangerine Peel Reheated by Stir-Frying, Steaming and Microwave"

_foods, 2025, doi:10.3390/foods14173017_

Round 1
Reviewer 1 Report
Comments and Suggestions for Authors
Comments to Author:
The paper assessed the effects of different reheating methods on the moisture content, lipid oxidation and flavour characteristics of beef with tangerine peel – a well-known traditional meat product in China.
The paper was interesting to read as it deals with how different reheating methods affect flavour profile. However, the authors need to make some improvements (as pointed out under Comments to consider).
The paper's strength lies in its application of methods and findings on the effects of reheating. However, the rationale for the study should be stronger, highlighting its importance/significance and how other researchers could benefit from the findings of this study.
Comments to consider:
Title (L1-3)
The paper has a suitable title, however, ‘with tangerine peel’ can cause some confusion as it is not clear how the peel was combined with the meat or what other ingredients are present. Perhaps the authors could make this clear.
Abstract (L19-35)
I would suggest that the authors start the abstract with 1-2 sentences to give background and the problem statement. What is this study of importance? Also, they need to conclude the abstract with the significance of the study – the ‘so what?’
L31-32: I The sentence ‘the principal component analysis (PCA) revealed that there were significant aroma differences among the four reheated samples’ should be rewritten as PCA cannot reveal significant differences. It is used to visualize data and groupings.
Keywords (L36)
Consider including words which reflect terms relevant to the content but not present in the title of the paper -replace ‘Beef with tangerine peel;’. Consider arranging alphabetically.
Introduction (L38-83)
L42: I would suggest the authors to provide more information about this product.
L52: The English language needs revision and improvement. … and the results indicated that… Is it ‘fishes’ (which does not make sense) or ‘dishes’.
L59: ‘researches’ should be ‘studies’. I would suggest the authors to check the rest of the paper and improve the language.
The authors sufficiently describe the aim of the study and the novelty. However, I think that the rationale for the study should be stronger. Why is it important? What is the significance of the work? How do other researchers benefit from the findings of this study?
Materials and methods (L84-201)
L95: It is not clear how many batches of the dish were prepared? I would expect several batches to have been prepared to improve the statistical power.
How did the authors ensure that the dishes were balanced regarding all the ingredients used for the different treatments? For instance, that one dish did not contain more rice than the other? Also, I do not see any description about the number of samples used.
Why was the sample not analysed before applying reheating?
L123: The authors refer to ‘beef samples’ – it is not clear if only beef or other ingredients (as part of the dish) were also analysed.
L154: The authors refer to ‘beef with tangerine peel samples’ – it is not clear what this sample entails regarding ingredients.
The authors refer to flavor components, yet the methods measure volatile organic compounds – which I think is a more fitting term.
L190: Descriptive sensory analysis (effective testing) cannot be used to measure ‘dislike’ and ‘like very much’. These are affective responses, best measured through other sensory evaluation methods like hedonic testing. This is a scientific flaw of the study.
The statistical analysis is very brief and could be more descriptive.
Results and Discussion (L202-465)
L221: Could the authors indicate the number of samples used in the caption of the figure? The same holds for the other figures/tables in the paper.
L224: The section title refers to ‘volatile compounds’ but in the section writes ‘flavor components’. I would recommend consistency with the use of terms.
L258: The authors speculate about the source of the volatiles but did not measure fatty acids in the study.
L358: It is not clear why SF was taken as the reference.
L449: The problem I have with this section is that descriptive sensory analysis cannot be used to test overall acceptability. Which is the only ‘attribute’ that differed significantly.
Although results were discussed fine separately, I missed the integration of the results. The panel could not distinguish any sensory differences, yet the analytical methods show differences, what is the significance of this?
Conclusions (L466-477)
I am missing the ‘so what’ of the study.
L477: The authors state ‘reheating treatments had a significant influence on the quality of beef’ yet found no sensory differences on the attributes that the panel can in principle measure… how is this conclusion justified?
Comments on the Quality of English LanguageEnglish Level: The English language is generally appropriate and understandable, but there are still some grammatical errors that can be improved.
Reviewer 2 Report
Comments and Suggestions for Authors
Line 143 – Give details of the previous method
Line 177 – how the supernatant was filtered?
Line 208 – 209 – To compare de moisture content, all the samples must be heated with open package or closed package. You can’t say that the heating has changed the moisture content.
When discussing about the TBAR, you can’t compare the results of a sealed environment to an open one. All samples must be at on environment or closed.
About Table 1, which of these compounds were found in the meat immediately after preparation.
Line 236 – “which might be due to the differences in heat transfer mechanisms”. What is the difference? Explain.
What is the effect of the higher concentration of hexanal, octanal and nonanal in the reheated samples?
What is the importance of having the highest hydrocarbons content?
In Topic 3.2 you should discuss the importance of the compounds found in the samples, why they are different and the impact of the difference in the flavours. In a general way, it is just presented the major constituents.
The presented work contains a large amount of data, but requires a better discussion of the results, highlighting the importance of each of the topics discussed. Furthermore, the work lacks a control sample, which would be the meat immediately after cooking, making it impossible to confirm that any changes occurred during heating.
Reviewer 3 Report
Comments and Suggestions for Authors
Foods
Manuscript Draft
Manuscript Number: 3788072
Title: Comparative Analysis of Flavor Quality of Beef with Tangerine Peel Reheated by Stir-frying, Boiling, Steaming and Microwave
Article Type: Research article
General Comments on MDPI Questions that Reviewers must answer:
- Is the manuscript clear, relevant, and presented in well?
This manuscript’s writing and structure needs to be improved. It is potentially relevant to the field since it uses evaluates four different ways of re-heating a popular beef dish in Sichuan, China. However, it is not clear how relevant the results are outside of this region in China. Given the potential contribution of this research, this manuscript has potential but needs significant improvement in order to be published in MDPI Foods. Please make the following TEN general substantive edits:
1) The Introduction is too short. Please provide more background on the significance and relevance of the dish studied. Why is it so important to successfully re-heat this dish as well as others? Why is so much food produced that re-heating is required? Please add a second paragraph on this in the Introduction section that provides more context on this. Any other additional contextual writing in the Introduction would be welcome due to how short this section is. What missing context do you need to provide the reader to better understand the results?
2) In the last paragraph of the Introduction section please more clearly state the goal(s) and then the objective(s) of the research. Literally use the word goal and objectives. What is the big picture goal of the study? Is this the same thing as the “aim” stated in the first sentence? What were the objective(s) of the study? Please write these out in the paragraph.
3) The writing style in many places has a tendency to “list” points, data, attributes, etc. with no stylistic variation in the writing. Please work with a native English speaker/editor to make improvements regarding this as well as other grammatical corrections throughout the manuscript.
4) There is an overuse of abbreviations throughout the manuscript. There should be NO abbreviations used in the Abstract, keywords, headers for major section, sub-sections, etc., as well as no abbreviations used in captions for figures and tables and the Conclusion section. Abbreviations should only be used if there is repeated use of the abbreviated term over and over again. It is better to “paraphrase” than overuse abbreviations. The reader needs to search the writing or footnotes to understand what the abbreviations stand for.
5) Please define abbreviations at the start of each major section. The general reading order of journal articles is not start to finish. Please make sure to define what abbreviations stand for at the start of each major section (i.e., Introduction, Methods, Results, Discussion).
6) Table and figures should be designed so that they are “stand-alone” in terms of understanding by the reader.
7) Please add a Discussion section with the following two subsections 4.1. Contrasts to Past Research and 4.2. Research Implications. The 4.1. Contrasts to Past Research sub-section describes how the results of the research compare to past studies. Note that more literature will need to be cited. For the 4.2. Research Implications sub-section, please write about the key implications of your research results. Why are your research results important and what do the results imply more broadly?
8) Paragraphs by definition are a minimum of 3 sentences (1 topic sentence followed by a minimum of 2 supporting sentences). Please correct paragraphs with only one sentence or two sentences by either breaking up sentences into shorter sentences or by adding more supporting sentences.
9) Please write out what SF, BR, SR, and MR stands for everywhere in the manuscript. This includes in figures and tables. This will reduce the length of the footnotes and make the figures and tables easier to understand just by looking at them.
10) Please increase the number of citations to at least 50 total. Since the Discussion section is being added, this will be where most of the additional citations are located.
Please also make the following TWENTY-FIVE minor edits:
1) Need to add phone number on L16.
2) Write out all abbreviations in the Abstract and “paraphrase” so there is no repetition.
3) The keywords need to be in alphabetical order with no abbreviations used.
4) Sub-sub-section headers do not capitalize all but the first word. For example on L95, change to: 2.2.1. Preparation of beef with tangerine peel
5) Change writing style on L111-117 from (#): listing to writing as regular sentences.
6) On L122, delete (TBARS) since this is redundant.
7) Write out what abbreviations stand for on L134.
8) Delete (OAV) on L148.
9) Write out what abbreviations stand for on L153.
10) Add another supporting sentence to L192-195.
11) Add another supporting sentence to L197-201.
12) Change L202 to Results
13) On L203, write as Oxidation
14) Everywhere in the manuscript, please make sure there is a blank space on either side of all +/- symbols such as on L207, etc. and in all tables.
15) On L224, write out what abbreviation stands for and reduce length of this sub-header.
16) In Table 1, etc. please avoid using the “/” symbol for “not detected”…why not just use: nd
17) On L289, write out what abbreviation stands for and reduce length of this sub-header.
18) Writing on L290-318 is just “listing” technical jargon. Please write out in general terms which should reduce the length of the paragraph. Same for L373-390.
19) On L342, write out what abbreviation stands for and reduce length of this sub-header.
20) Sub-header on L405-406 is too long. Same for L432-433 and L449-450.
21) In Table 3, for the 1st column please write out what the amino acids are without using abbreviations. There is enough space to do so and it will improve understanding.
22) In Table 3, Table 4, and elsewhere, add a blank space on either side of the +/- symbols.
23) Y-axis labels for Figure 4 pictures are so small they are very difficult to read.
24) Please add a couple of sentences at the end of the Conclusions section’s paragraph on what future research can do to improve upon the current work.
25) In the References section for citations, please a) both volume or volume(issue) need to be in italics with no space before the open parentheses symbol, b) for all page ranges, please use the longer endash symbol and not the shorter hyphen (refer back to the Word template), and c) please also add DOI links for all references at the end of the citation.
- Are the cited references published within the last 5 years or so that they are relevant? Are there an excessive number of self-citations?
There are about 27 out of the 36 cited references that have been published since 2019. All citations appear to be relevant to the topic. Please increase the number of citations to 50 or more. There are no excessive self-citations.
- Is the research scientifically sound and the experimental design appropriate?
The research is scientifically sound. The analyses are appropriate.
- Are the manuscript’s results reproducible based on the methods?
The results can be reproduced based on what is written in the methods.
- Are the figures/tables/data appropriate?
The figures and tables require minor edits that have already been specified.
- Are the conclusions consistent with the evidence and the arguments presented?
Please add a couple of sentences to end of the Conclusions section’s paragraph summarizing implications of the results based on the newly added Discussion section. Please also add a couple of sentences on how future research can improve upon the current work.
- Please evaluate the data availability and conflicts of interest statements.
The data availability statement and conflict of interest statement are OK. Please add an Acknowledgements section. Please refer to the Word template on the MDPI Foods website regarding these Back Matter sections.
Comments on the Quality of English LanguagePlease have a native English speaker review and edit the writing.
Round 2
Reviewer 2 Report
Comments and Suggestions for Authors
The manuscript has undergone significant improvements in its discussion of the results. However, the text continues to focus on whether heating was performed in an open environment or not. Despite the authors' response, it remains unclear how the process was performed, and if heating was performed in different ways, this is a serious methodological flaw. This section needs to be revised again so that the results are truly comparable.
Reviewer 3 Report
Comments and Suggestions for Authors
Foods
Manuscript Draft
Manuscript Number: 3788072
Title: Comparative Analysis of Flavor Quality of Beef with Tangerine Peel Reheated by Stir-frying, Boiling, Steaming and Microwave
Article Type: Research article
Thank-you for making requested edits. Please make the following SIX minor edits:
1) On L26, delete (OAVs).
2) Delete OAVs on L29 and L30 and instead write out.
3) On L33, delete (GC-IMS) and (PCA) as these abbreviations are redundant. In general. abbreviations are NOT used in the Abstract. Write out all abbreviations in the Abstract and “paraphrase” so there is no repetition.
4) For a, b, c, d lower case letters denoting significant differences in tables, please do NOT format as superscript. Keep the font formatting normal making sure there is a blank space between the lower case letter and the preceding +/- range.
5) In Table 2, add a footnote defining what nd stands for (not detected). The footnote number will be superscripted after all nd’s in the table so should look like: nd1
6) In the tables, if a number is +/- 0.00 is that literally mean zero so there is no variation? Could that be written instead as +/- <0.01?
